# Probabilistic Inference with Generating Functions for Poisson Latent Variable Models

**Kevin Winner**[1] **and Daniel Sheldon**[1,2]
`{kwinner,sheldon}@cs.umass.edu`

[1] College of Information and Computer Sciences, University of Massachusetts Amherst
[2] Department of Computer Science, Mount Holyoke College

## Abstract

Graphical models with latent count variables arise in a number of fields. Standard exact inference techniques such as variable elimination and belief propagation do not apply to these models because the latent variables have countably infinite support. As a result, approximations such as truncation or MCMC are employed. We present the first *exact* inference algorithms for a class of models with latent count variables by developing a novel representation of countably infinite factors as probability generating functions, and then performing variable elimination with generating functions. Our approach is exact, runs in pseudo-polynomial time, and is much faster than existing approximate techniques. It leads to better parameter estimates for problems in population ecology by avoiding error introduced by approximate likelihood computations.

## 1   Introduction

A key reason for the success of graphical models is the existence of fast algorithms that exploit the graph structure to perform inference, such as Pearl's belief propagation [19] and related propagation algorithms [13, 16, 23] (which we refer to collectively as "message passing" algorithms), and variable elimination [27]. For models with a simple enough graph structure, these algorithms can compute marginal probabilities exponentially faster than direct summation.

However, these fast exact inference methods apply only to a relatively small class of models—those for which the basic operations of marginalization, conditioning, and multiplication of constituent factors can be done efficiently. In most cases, this means that the user is limited to models where the variables are either discrete (and finite) or Gaussian, or they must resort to some approximate form of inference. Why are Gaussian and discrete models tractable while others are not? The key issue is one of *representation*. If we start with factors that are all discrete or all Gaussian, then: (1) factors can be represented exactly and compactly, (2) conditioning, marginalization, and multiplication can be done efficiently in the compact representation, and (3) each operation produces new factors of the same type, so they can also be represented exactly and compactly.

Many models fail the restriction of being discrete or Gaussian even though they are qualitatively "easy". The goal of this paper is to expand the class of models amenable to fast exact inference by developing and exploiting a novel representation for factors with properties similar to the three above. In particular, we investigate models with *latent count variables*, and we develop techniques to represent and manipulate factors using *probability generating functions*.

Figure 1 provides a simple example to illustrate the main ideas. It shows a model that is commonly used to interpret field surveys in ecology, where it is known as an *N-mixture* model [22]. The latent variable $n \sim \text{Poisson}(\lambda)$ represents the unknown number of individual animals at a given site. Repeated surveys are conducted at the site during which the observer detects each individual with

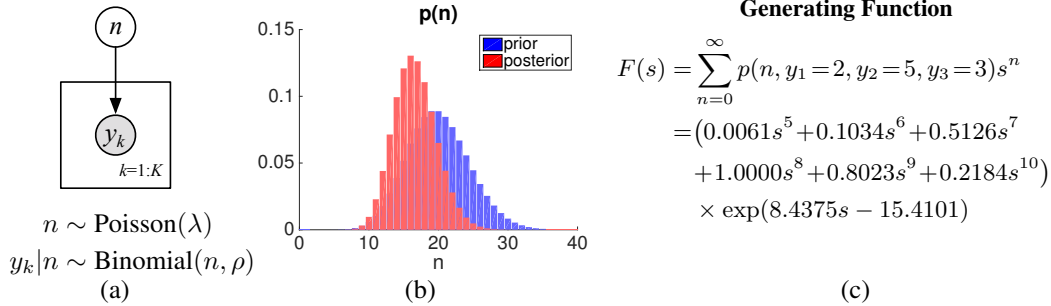

Figure 1: The N-mixture model [22] is a simple model with a Poisson latent variable for which no exact inference algorithm is known: (a) the model, (b) the prior and posterior for $\lambda = 20$, $\rho = 0.25, y_1 = 2, y_2 = 5, y_3 = 3$, (c) a closed form representation of the generating function of the unnormalized posterior, which is a compact and exact description of the posterior.

probability $\rho$, so each observation $y_k$ is $\mathrm{Binomial}(n, \rho)$. From these observations (usually across many sites with shared $\lambda$), the scientist wishes to infer $n$ and fit $\lambda$ and $\rho$.

This model is very simple: all variables are marginally Poisson, and the unnormalized posterior has a simple form (e.g., see Figure 1b). However, until recently, there was no known algorithm to exactly compute the likelihood $p(y_{1:K})$. The naive way is to sum the unnormalized posterior $p(n, y_1, \ldots, y_K)$ over all possible values of $n$. However, $n$ has a countably infinite support, so this is not possible. In practice, users of this and related models truncate the infinite sum at a finite value [22]. A recent paper developed an exact algorithm for the N-mixture model, but one with running time that is exponential in $K$ [8]. For a much broader class of models with **Poisson latent variables** [5, 7, 11, 15, 28], there are *no known exact inference algorithms*. Current methods either truncate the support [5, 7, 11], which is slow (e.g., see [4]) and interacts poorly with parameter estimation [6, 8], or use MCMC [15, 28], which is slow and for which convergence is hard to assess. The key difficulty with these models is that we lack *finite and computationally tractable representations* of factors over variables with a countably infinite support, such as the posterior distribution in the N-mixture model, or intermediate factors in exact inference algorithms.

The main contribution of this paper is to develop *compact* and *exact* representations of countably infinite factors using probability generating functions (PGFs) and to show how to perform variable elimination in the domain of generating functions. We provide the first exact pseudo-polynomial time inference algorithms (i.e., polynomial in the magnitude of the observed variables) for a class of Poisson latent variable models, including the N-mixture model and a more general class of Poisson HMMs. For example, the generating function of the unnormalized N-mixture posterior is shown in Figure 1c, from which we can *efficiently* recover the likelihood $p(y_1 = 2, y_2 = 5, y_3 = 3) = F(1) = 0.0025$. For Poisson HMMs, we first develop a PGF-based forward algorithm to compute the likelihood, which enables efficient parameter estimation. We then develop a "tail elimination" approach to compute posterior marginals. Experiments show that our exact algorithms are much faster than existing approximate approaches, and lead to better parameter estimation.

**Related work.** Several previous works have used factor transformations for inference. Bickson and Guestrin [2] show how to perform inference in the space of characteristic functions (see also [17]) for a certain class of factor graphs. Xue et al. [26] perform variable elimination in discrete models using Walsh-Hadamard transforms. Jha et al. [14] use generating functions (over finite domains) to compute the partition function of Markov logic networks. McKenzie [18] describes the use of PGFs in discrete time series models, which are related to our models except they are fully observed, and thus require no inference.

## 2   The Poisson Hidden Markov Model

Although our PGF-based approaches will apply more broadly, the primary focus of our work is a Poisson hidden Markov model (HMM) that captures a number of models from different disciplines. To describe the model, we first introduce notation for an operation called *binomial thinning* [24].

Write $z = \rho \circ n$ to mean that $z|n \sim \text{Binomial}(n, \rho)$, i.e., $z$ is the result of "thinning" the $n$ individuals so that each remains with probability $\rho$. The Poisson HMM model is given by:

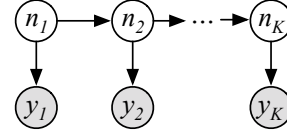

$$n_k = \text{Poisson}(\lambda_k) + \delta_{k-1} \circ n_{k-1}, \quad y_k = \rho_k \circ n_k.$$

for $k \geq 1$, with the initialization condition $n_0 = 0$. The variables $n_1, \ldots, n_K$ describe the size of a population at sampling times $t_1 <$

Figure 2: Poisson HMM

$t_2 < \ldots < t_K$. At time $t_k$, the population consists of a $\text{Poisson}(\lambda_k)$ number of new arrivals, plus $\delta_{k-1} \circ n_{k-1}$ survivors from the previous time step (each individual survives with probability $\delta_k$). A noisy count $y_k = \rho_k \circ n_k$ is made of the population at time $t_k$, where $\rho_k$ is the detection probability of each individual. This model is broadly applicable. It models situations where individuals arrive in an iid fashion, and the time they remain is "memoryless". Versions of this model are used in ecology to model surveys of "open populations" (individuals arrive and depart over time) [7] and the timing and abundance of insect populations [12, 25, 29], and it also capture models from queueing theory [9] and generic time series models for count data [1, 18].

**Existing approaches.** Two classes of methods have been applied for inference in Poisson HMMs and related models. The first is to truncate the support of the Poisson variables at a large but finite value $N_{\max}$ [5, 7, 11, 22]. Then, for example, the Poisson HMM reduces to a standard discrete HMM. This is unsatisfactory because it is slow (a smart implementation that uses the fast Fourier transform takes time $O(K N_{\max}^2 \log N_{\max})$), and the choice of $N_{\max}$ is intertwined with the unknown Poisson parameters $\lambda_k$, so the approximation interacts poorly with parameter estimation [6, 8]. The second class of approximate methods that has been applied to these problems is MCMC [28]. This is undesirable because it is also slow, and because the problem has a simple structure that should admit fast algorithms.

## 3   Variable Elimination with Generating Functions

Our approach to inference in Poisson HMMs will be to implement the same abstract set of operations as variable elimination, but using a representation based on probability generating functions. Because variable elimination will produce intermediate factors on larger sets of variables, and to highlight the ability of our methods to generalize to a larger class of models, we first abstract from the Poisson HMM to introduce notation general for graphical models with multivariate factors, and their corresponding multivariate generating functions.

**Factors.** Let $x = (x_1, \ldots, x_d)$ be a vector of nonnegative integer-valued random variables where $x_i \in \mathcal{X}_i \subseteq \mathbb{Z}_{\geq 0}$. The set $\mathcal{X}_i$ may be finite (e.g., to model binary or finite discrete variables), but we assume without loss of generality that $\mathcal{X}_i = \mathbb{Z}_{\geq 0}$ for all $i$ by defining factors to take value zero for integers outside of $\mathcal{X}_i$. For any set $\alpha \subseteq \{1, \ldots, d\}$, define the subvector $x_\alpha := (x_i, i \in \alpha)$. We consider probability models of the form $p(x) = \frac{1}{Z} \prod_{\alpha \in \mathcal{A}} \psi_\alpha(x_\alpha)$, where $Z$ is a normalization constant and $\{\psi_\alpha\}$ is a set of factors $\psi_\alpha : \mathbb{Z}_{\geq 0} \to \mathbb{R}^+$ indexed by subsets $\alpha \subseteq \{1, \ldots, d\}$ in a collection $\mathcal{A}$.

**Generating Functions.** A general factor $\psi_\alpha$ on integer-valued variables cannot be finitely represented. We instead use the formalization of probability generating functions (PGFs). Let $s = (s_1, \ldots, s_d)$ be a vector of indeterminates corresponding to the random variables $x$. The joint PGF of a factor $\psi_\alpha$ is

$$F_\alpha(s_\alpha) = \sum_{x_\alpha} \psi_\alpha(x_\alpha) \cdot \prod_{i \in \alpha} s_i^{x_i} = \sum_{x_\alpha} \psi_\alpha(x_\alpha) \cdot s_\alpha^{x_\alpha}.$$

Here, for two vectors $a$ and $b$ with the same index set $\mathcal{I}$, we have defined $a^b = \prod_{i \in \mathcal{I}} a_i^{b_i}$. The sum is over all vectors $x_\alpha$ of non-negative integers.

Univariate PGFs of the form $F(s) = \sum_{x=0}^{\infty} \Pr(X = x) s^x = \mathbb{E}[s^X]$, where $X$ is a nonnegative integer-valued random variable, are widely used in probability and statistics [3, 21], and have a number of nice properties. A PGF uniquely encodes the distribution of $X$, and there are formulas to recover moments and entries of the the probability mass function from the PGF. Most common distributions have closed-form PGFs, e.g., $F(s) = \exp\{\lambda(s-1)\}$ when $X \sim \text{Poisson}(\lambda)$. Similarly, the joint PGF $F_\alpha$ uniquely encodes the factor $\psi_\alpha$, and we will develop a set of useful operations on joint PGFs. Note that we abuse terminology slightly by referring to the generating function of the

factor $\psi_\alpha$ as a *probability* generating function; however, it is consistent with the view of $\psi_\alpha$ as an unnormalized probability distribution.

## 3.1 Operations on Generating Functions

Our goal is to perform variable elimination using factors represented as PGFs. To do this, the basic operations we need to support are are multiplication, marginalization, and "entering evidence" into factors (reducing the factor by fixing the value of one variable). In this section we state a number of results about PGFs that show how to perform such operations. For the most part, these are either well known or variations on well known facts about PGFs (e.g., see [10], Chapters 11, 12). All proofs can be found in the supplementary material.

First, we see that marginalization of factors is very easy in the PGF domain:

**Proposition 1** (Marginalization). *Let $\psi_{\alpha\setminus i}(x_{\alpha\setminus i}) := \sum_{x_i \in \mathcal{X}_i} \psi_\alpha(x_{\alpha\setminus i}, x_i)$ be the factor obtained from marginalizing $i$ out of $\psi_\alpha$. The joint PGF of $\psi_{\alpha\setminus i}$ is $F_{\alpha\setminus i}(s_{\alpha\setminus i}) = F_\alpha(s_{\alpha\setminus i}, 1)$. The normalization constant $\sum_{x_\alpha} \psi_\alpha(x_\alpha)$ is equal to $F_\alpha(1, \ldots, 1)$.*

Entering evidence is also straightforward:

**Proposition 2** (Evidence). *Let $\psi_{\alpha\setminus i}(x_{\alpha\setminus i}) := \psi_\alpha(x_{\alpha\setminus i}, a)$ be the factor resulting from observing the value $x_i = a$ in $\psi_\alpha$. The joint PGF of $\psi_{\alpha\setminus i}$ is $F_{\alpha\setminus i}(s_{\alpha\setminus i}) = \frac{1}{a!} \frac{\partial^a}{\partial s_i^a} F_\alpha(s_\alpha)\big|_{s_i=0}$.*

Multiplication in the PGF domain—i.e., computing the PGF of the product $\psi_\alpha(x_\alpha)\psi_\beta(x_\beta)$ of two factors $\psi_\alpha$ and $\psi_\beta$—is not straightforward in general. However, for certain types of factors, multiplication is possible. We give two cases.

**Proposition 3** (Multiplication: Binomial thinning). *Let $\psi_{\alpha\cup j}(x_\alpha, x_j) = \psi_\alpha(x_\alpha) \cdot \mathrm{Binomial}(x_j|x_i, \rho)$ be the factor resulting from expanding $\psi_\alpha$ to introduce a thinned variable $x_j := \rho \circ x_i$, where $i \in \alpha$ and $j \notin \alpha$. The joint PGF of $\psi_{\alpha\cup j}$ is $F_{\alpha\cup j}(s_\alpha, s_j) = F_\alpha(s_{\alpha\setminus i}, s_i(\rho s_j + 1 - \rho))$.*

**Proposition 4** (Multiplication: Addition of two variables). *Let $\psi_\gamma(x_\alpha, x_\beta, x_k) := \psi_\alpha(x_\alpha)\psi_\beta(x_\beta)\mathbb{I}\{x_k = x_i + x_j\}$ be the joint factor resulting from the introduction of a new variable $x_k = x_i + x_j$, where $i \in \alpha$, $j \in \beta$, $k \notin \alpha \cup \beta$, $\gamma := \alpha \cup \beta \cup \{k\}$. The joint PGF of $\psi_\gamma$ is $F_\gamma(s_\alpha, s_\beta, s_k) = F_\alpha(s_{\alpha\setminus i}, s_k s_i)F_\beta(s_{\beta\setminus j}, s_k s_j)$.*

The four basic operations above are enough to perform variable elimination on a large set of models. In practice, it is useful to introduce additional operations that combine two of the above operations.

**Proposition 5** (Thin then observe). *Let $\psi'_\alpha(x_\alpha) := \psi_\alpha(x_\alpha) \cdot \mathrm{Binomial}(a|x_i, \rho)$ be the factor resulting from observing the thinned variable $\rho \circ x_i = a$ for $i \in \alpha$. The joint PGF of $\psi'_\alpha$ is $F'_\alpha(s_\alpha) = \frac{1}{a!}(s_i\rho)^a \frac{\partial^a}{\partial t_i^a} F_\alpha(s_{\alpha\setminus i}, t_i)\big|_{t_i = s_i(1-\rho)}$.*

**Proposition 6** (Thin then marginalize). *Let $\psi_{(\alpha\setminus i)\cup j}(x_{\alpha\setminus i}, x_j) := \sum_{x_i} \psi_\alpha(x_\alpha) \cdot \mathrm{Binomial}(x_j|x_i, \rho)$ be the factor resulting from introducing $x_j := \rho \circ x_i$ and then marginalizing $x_i$ for $i \in \alpha, j \notin \alpha$. The joint PGF of $\psi_{(\alpha\setminus i)\cup j}$ is $F_{(\alpha\setminus i)\cup j}(s_{\alpha\setminus i}, s_j) = F_\alpha(s_{\alpha\setminus i}, \rho s_j + 1 - \rho)$.*

**Proposition 7** (Add then marginalize). *Let $\psi_\gamma(x_{\alpha\setminus i}, x_{\beta\setminus j}, x_k) := \sum_{x_i, x_j} \psi_\alpha(x_\alpha)\psi_\beta(x_\beta)\mathbb{I}\{x_k = x_i + x_j\}$ be the factor resulting from the deterministic addition $x_i + x_j = x_k$ followed by marginalization of $x_i$ and $x_j$, where $i \in \alpha, j \in \beta, k \notin \alpha \cup \beta, \gamma := (\alpha \setminus i) \cup (\beta \setminus j) \cup \{k\}$. The joint PGF of $\psi_\gamma$ is $F_\gamma(s_{\alpha\setminus i}, s_{\beta\setminus j}, s_k) = F_\alpha(s_{\alpha\setminus i}, s_k)F_\beta(s_{\beta\setminus j}, s_k)$.*

## 3.2 The PGF-Forward Algorithm for Poisson HMMs

We now use the operations from the previous section to implement the forward algorithm for Poisson HMMs in the domain of PGFs. The forward algorithm is an instance of variable elimination, but in HMMs is more easily described using the following recurrence for the joint probability $p(n_k, y_{1:k})$:

$$\underbrace{p(n_k, y_{1:k})}_{\alpha_k(n_k)} = \sum_{n_{k-1}} \underbrace{p(n_{k-1}, y_{1:k-1})}_{\alpha_{k-1}(n_{k-1})} p(n_k|n_{k-1})p(y_k|n_k)$$

We can compute the "forward messages" $\alpha_k(n_k) := p(n_k, y_{1:k})$ in a sequential forward pass, assuming it is possible to enumerate all possible values of $n_k$ to store the messages and compute the recurrence. In our case, $n_k$ can take on an infinite number of values, so this is not possible.

| **Algorithm 1** FORWARD | **Algorithm 2** PGF-FORWARD |
|---|---|
| 1: $\psi_1(z_1) := \mathbb{I}\{z_1 = 0\}$ | 1: $\Psi_1(s) := 1$ |
| 2: **for** $k = 1$ to $K$ **do** | 2: **for** $k = 1$ to $K$ **do** |
| 3: $\quad \gamma_k(n_k) := \sum\limits_{z_k, m_k} \psi_k(z_k)p(m_k)\mathbb{I}\{n_k = z_k + m_k\}$ | 3: $\quad \Gamma_k(s) := \Psi_k(s) \cdot \exp\{\lambda_k(s-1)\}$ |
| 4: $\quad \alpha_k(n_k) := \gamma_k(n_k)p(y_k \mid n_k)$ | 4: $\quad A_k(s) := \frac{1}{y_k!}(s\rho_k)^{y_k}\Gamma_k^{(y_k)}\big(s(1-\rho_k)\big)$ |
| 5: $\quad$ **if** $k < K$ **then** | 5: $\quad$ **if** $k < K$ **then** |
| 6: $\qquad \psi_{k+1}(z_{k+1}) := \sum_{n_k} \alpha_k(n_k)p(z_{k+1} \mid n_k)$ | 6: $\qquad \Psi_{k+1}(s) := A_k\big(\delta_k s + 1 - \delta_k\big)$ |
| 7: $\quad$ **end if** | 7: $\quad$ **end if** |
| 8: **end for** | 8: **end for** |

We proceed instead using generating functions. To apply the operations from the previous section, it is useful to instantiate explicit random variables $m_k$ and $z_k$ for the number of new arrivals in step $k$ and survivors from step $k-1$, respectively, to get the model (see Figure 3):

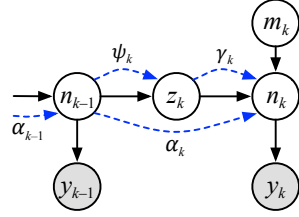

$$m_k \sim \text{Poisson}(\lambda_k), \qquad z_k = \delta_{k-1} \circ n_{k-1},$$
$$n_k = m_k + z_k, \qquad y_k = \rho_k \circ n_k.$$

Figure 3: Expanded model.

We can now expand the recurrence for $\alpha_k(n_k)$ as:

$$\alpha_k(n_k) = p(y_k|n_k) \underbrace{\sum_{m_k=0}^{\infty} \sum_{z_k=0}^{\infty} p(m_k)p(n_k|z_k, m_k) \overbrace{\sum_{n_{k-1}=0}^{\infty} \alpha_{k-1}(n_{k-1})p(z_k|n_{k-1})}^{\psi_k(z_k)}}_{\gamma_k(n_k)} \quad (1)$$

We have introduced the intermediate factors $\psi_k(z_k)$ and $\gamma_k(n_k)$ to clarify the implementation.

FORWARD (Algorithm 1) is a dynamic programming algorithm based on this recurrence to compute the $\alpha_k$ messages for all $k$. However, it cannot be implemented due to the infinite sums. PGF-FORWARD (Algorithm 2) instead performs the same operations in the domain of generating functions—$\Psi_k$, $\Gamma_k$, and $A_k$ are the PGFs of $\psi_k$, $\gamma_k$, and $\alpha_k$, respectively. Each line in PGF-FORWARD implements the operation in the corresponding line of FORWARD using the operations given in Section 3.1. In Line 1, $\Psi_1(s) = \sum_{z_1} \psi_1(z_1)s^{z_1} = 1$ is the PGF of $\psi_1$. Line 3 uses "Add then marginalize" (Proposition 7) combined with the fact that the Poisson PGF for $m_k$ is $\exp\{\lambda_k(s-1)\}$. Line 4 uses "Thin then observe" (Proposition 5), and Line 6 uses "Thin then marginalize" (Proposition 6).

**Implementation and Complexity.** The PGF-FORWARD algorithm as stated is symbolic. It remains to see how it can be implemented efficiently. For this, we need to respresent and manipulate the PGFs in the algorithm efficiently. We do so based on the following result:

**Theorem 1.** *All PGFs in the* PGF-FORWARD *algorithm have the form* $f(s)\exp\{as + b\}$ *where $f$ is a polynomial with degree at most* $Y = \sum_k y_k$.

*Proof.* We verify the invariant inductively. It is clearly satisfied in Line 1 of PGF-FORWARD ($f(s) = 1, a = b = 0$). We check that it is preserved for each operation within the loop. In Line 3, suppose $\Psi_k(s) = f(s)\exp\{as + b\}$. Then $\Gamma_k(s) = f(s)\exp\{(a + \lambda_k)s + (b - \lambda_k)\}$ has the desired form.

In Line 4, assume that $\Gamma_k(s) = f(s)\exp\{as + b\}$. Then one can verify by taking the $y_k$th derivative of $\Gamma_k(s)$ that $A_k(s)$ is given by:

$$A_k(s) = (a\rho_k)^{y_k} \cdot \left(s^{y_k} \sum_{\ell=0}^{y_k} \frac{f^{(\ell)}(s(1-\rho_k))}{a^{\ell}\ell!(y_k - \ell)!}\right) \cdot \exp\{a(1 - \rho_k)s + b\}$$

The scalar $(a\rho)^{y_k}$ can be combined with the polynomial coefficients or the scalar $\exp(b)$ in the exponential. The second term is a polynomial of degree $y_k + \deg(f)$. The third term has the form $\exp\{a's + b'\}$. Therefore, in Line 4, $A_k(s)$ has the desired form, and the degree of the polynomial part of the representation increases by $y_k$.

In Line 6, suppose $A_k(s) = f(s) \exp\{as + b\}$. Then $\Psi_{k+1}(s) = g(s) \exp\{a\delta_k s + (b + a(1 - \delta_k))\}$, where $g(s)$ is the composition of $f$ with the affine function $\delta_k s + 1 - \delta_k$, so $g$ is a polynomial of the same degree as $f$. Therefore, $\Psi_{k+1}(s)$ has the desired form.

We have shown that each PGF retains the desired form, and the degree of the polynomial is initially zero and increases by $y_k$ each time through the loop, so it is always bounded by $Y = \sum_k y_k$. $\qquad\square$

The important consequence of Theorem 1 is that we can represent and manipulate PGFs in PGF-FORWARD by storing at most $Y$ coefficients for the polynomial $f$ plus the scalars $a$ and $b$. An efficient implementation based on this principle and the proof of the previous theorem is given in the supplementary material.

**Theorem 2.** *The running time of* PGF-FORWARD *for Poisson HMMs is* $\mathcal{O}(KY^2)$.

### 3.3 Computing Marginals by Tail Elimination

PGF-FORWARD allows us to efficiently compute the likelihood in a Poisson HMM. We would also like to compute posterior marginals, the standard approach for which is the forward-backward algorithm [20]. A natural question is whether there is an efficient PGF implementation of the backward algorithm for Poisson HMMs. While we were able to derive this algorithm symbolically, the functional form of the PGFs is more complex and we do not know of a polynomial-time implementation. Instead, we adopt a variable elimination approach that is less efficient in terms of the number of operations performed on factors ($\mathcal{O}(K^2)$ instead of $\mathcal{O}(K)$ to compute all posterior marginals) but with the significant advantage

---

**Algorithm 3** PGF-TAIL-ELIMINATE

**Output:** PGF of unnormalized marginal $p(n_i, y_{1:K})$
1: $\Phi_{i,i+1}(s,t) := A_i(s(\delta_i t + 1 - \delta_i))$
2: **for** $j = i + 1$ to $K$ **do**
3: $\quad H_{ij}(s,t) := \Phi_{ij}(s,t) \exp\{\lambda_k(t - 1)\}$
4: $\quad \Theta_{ij}(s,t) := \frac{1}{y_j!}(t\rho_j)^{y_j} \frac{\partial^{y_j} H_{ij}(s,u)}{\partial u^{y_j}}\Big|_{u = t(1 - \rho_j)}$
5: $\quad$ **if** $j < K$ **then**
6: $\qquad \Phi_{i,j+1}(s,t) := \Theta_{ij}(s, \delta_j t + 1 - \delta_j)$
7: $\quad$ **end if**
8: **end for**
9: **return** $\Theta_{iK}(s,1)$

---

that those operations are efficient. The key principle is to always eliminate predecessors before successors in the Poisson HMM. This allows us to apply operations similar to those in PGF-FORWARD.

Define $\theta_{ij}(n_i, n_j) := p(n_i, n_j, y_{1:j})$ for $j > i$. We can write a recurrence for $\theta_{ij}$ similar to Equation (1). For $j > i + 1$:

$$\theta_{ij}(n_i, n_j) = p(y_j | n_j) \sum_{m_j, z_j} p(m_j) p(n_j | z_j, m_j) \underbrace{\overbrace{\sum_{n_{j-1}} \theta_{i,j-1}(n_i, n_{j-1}) p(z_j | n_{j-1})}^{\phi_{ij}(n_i, z_j)}}_{\eta_{ij}(n_i, n_j)}.$$

We have again introduced intermediate factors, with probabilistic meanings $\phi_{ij}(n_i, z_j) = p(n_i, z_j, y_{1:j-1})$ and $\eta_{ij}(n_i, n_j) = p(n_i, n_j, y_{1:j-1})$.

PGF-TAIL-ELIMINATE (Algorithm 3) is a PGF-domain dynamic programming algorithm based on this recurrence to compute the PGFs of the $\theta_{ij}$ factors for all $j \in \{i + 1, \dots, K\}$. The non-PGF version of the algorithm appears in the supplementary material for comparison. We use $\Theta_{ij}$, $\Phi_{ij}$, and $H_{ij}$ to represent the joint PGFs of $\theta_{ij}$, $\phi_{ij}$, and $\eta_{ij}$, respectively. The algorithm can also be interpreted as variable elimination using the order $z_{i+1}, n_{i+1}, \dots, z_K, n_K$, after having already eliminated variables $n_{1:i-1}$ and $z_{1:i-1}$ in the forward algorithm, and therefore starting with the PGF of $\alpha_i(n_i)$. PGF-TAIL-ELIMINATE concludes by marginalizing $n_K$ from $\Theta_{iK}$ to obtain the PGF of the unnormalized posterior marginal $p(n_i, y_{1:K})$. Each line of PGF-TAIL-ELIMINATE uses the same operations given in Section 3.1. Line 1 uses "Binomial thinning" (Proposition 3), Line 3 uses "Add then marginalize" (Proposition 7), Line 4 uses "Thin then observe" (Proposition 5) and Line 6 uses "Thin then marginalize" (Proposition 6).

**Implementation and Complexity.** The considerations for implementating PGF-TAIL-ELIMINATE are similar to those of PGF-FORWARD, with the details being slightly more complex due to the larger factors. We state the main results here and include proofs and implementation details in the supplementary material.

**Theorem 3.** *All PGFs in the* PGF-TAIL-ELIMINATE *algorithm have the form* $f(s,t) \exp\{ast + bs + ct + d\}$ *where $f$ is a bivariate polynomial with maximum exponent most* $Y = \sum_k y_k$.

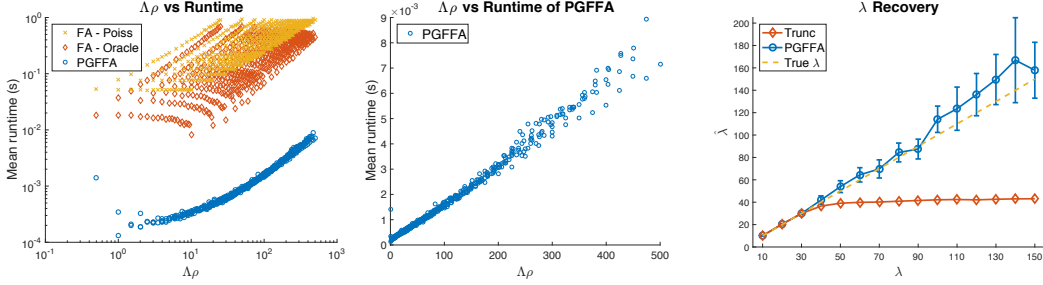

Figure 4: Runtime of PGF-FORWARD and truncated algorithm vs. $\Lambda\rho$. Left: log-log scale. Right: PGF-FORWARD only, linear scale.

Figure 5: Parameter estimation w/ PGF-FORWARD

**Theorem 4.** PGF-TAIL-ELIMINATE *can be implemented to run in time* $\mathcal{O}(Y^3(\log Y + K))$, *and the PGFs for all marginals can be computed in time* $\mathcal{O}(KY^3(\log Y + K))$.

### 3.4 Extracting Posterior Marginals and Moments

After computing the PGF of the posterior marginals, we wish to compute the actual probabilities and other quantities, such as the moments, of the posterior distribution. This can be done efficiently:

**Theorem 5.** *The PGF of the unnormalized posterior marginal* $p(n_i, y_{1:K})$ *has the form* $F(s) = f(s)\exp\{as + b\}$ *where* $f(s) = \sum_{j=0}^m c_j s^j$ *is a polynomial of degree* $m \leq Y$. *Given the parameters of the PGF, the posterior mean, the posterior variance, and an arbitrary entry of the posterior probability mass function can each be computed in* $\mathcal{O}(m) = \mathcal{O}(Y)$ *time as follows, where* $Z = f(1)\exp\{a + b\}$:

(i) $\mu := \mathbb{E}[n_i \mid y_{1:k}] = e^{a+b-\log Z} \sum_{j=0}^m (a + m)c_j$

(ii) $\sigma^2 := \text{Var}(n_i \mid y_{1:k}) = \mu - \mu^2 + e^{a+b-\log Z} \sum_{j=0}^m ((a+m)^2 - m)c_j$

(iii) $\Pr(n_i = \ell \mid y_{1:k}) = e^{b-\log Z} \sum_{j=0}^{\min\{m,\ell\}} c_j \frac{a^{\ell-i}}{(\ell-i)!}$

## 4 Experiments

We conducted experiments to demonstrate that our method is faster than standard approximate approaches for computing the likelihood in Poisson HMMs, that it leads to better parameter estimates, and to demonstrate the computation of posterior marginals on an ecological data set.

**Running time.** We compared the runtimes of PGF-FORWARD and the truncated forward algorithm, a standard method for Poisson HMMs in the ecology domain [7]. The runtime of our algorithm depends on the magnitude of the observed counts. The runtime of the truncated forward is very sensitive to the setting of the truncation parameter $N_{\max}$: smaller values are faster, but may underestimate the likelihood. Selecting $N_{\max}$ large enough to yield correct likelihoods but small enough to be fast is difficult [4, 6, 8]. We evaluated two strategies to select $N_{\max}$. The first is an oracle strategy, where we first searched for the smallest value of $N_{\max}$ for which the error in the likelihood is at most $0.001$, and then compared vs. the runtime for that value (excluding the search time). The second strategy, adapted from [8], is to set $N_{\max}$ such that the maximum discarded tail probability of the Poisson prior over any $n_k$ is less than $10^{-5}$.

To explore these issues we generated data from models with arrival rates $\lambda = \Lambda \times [0.0257, 0.1163, 0.2104, 0.1504, 0.0428]$ and survival rates $\delta = [0.2636, 0.2636, 0.2636, 0.2636]$ based on a model for insect populations [29]. We varied the overall population size parameter $\Lambda \in \{10, 20, \ldots, 100, 125, 150, \ldots, 500\}$, and detection probability $\rho \in \{0.05, 0.10, \ldots, 1.00\}$. For each parameter setting, we generated 25 data sets and recorded the runtime of both methods.

Figure 4 shows that PGF-FORWARD is 2–3 orders of magnitude faster than even the oracle truncated algorithm. The runtime is plotted against $\Lambda\rho \propto \mathbb{E}[Y]$, the primary parameter controlling the runtime of PGF-FORWARD. Empirically, the runtime depends linearly instead of quadratically, as predicted,

on the magnitude of observed counts—this is likely due to the implementation, which is dominated by loops that execute $\mathcal{O}(Y)$ times, with much faster vectorized $\mathcal{O}(Y)$ operations within the loops.

**Parameter Estimation.** We now examine the impact of exact vs. truncated likelihood computations on parameter estimation in the N-mixture model [22]. A well-known feature of this and related models is that it is usually easy to estimate the product of the population size parameter $\lambda$ and detection probability $\rho$, which determines the mean of the observed counts, but, without enough data, it is difficult to estimate both parameters accurately, especially as $\rho \to 0$ (e.g., see [8]). It was previously shown that truncating the likelihood can artificially suppress instances where the true maximum-likelihood estimates are infinite [8], a phenomenon that we also observed. We designed a different, simple, experiment to reveal another failure case of the truncated likelihood, which is avoided by our exact methods. In this case, the modeler is given observed counts over 50 time steps $(K = 50)$ at 20 iid locations. She selects a heuristic fixed value of $N_{\max}$ approximately 5 times the average observed count based on her belief that the detection probability is not too small and this will capture most of the probability mass.

To evaluate the accuracy of parameter estimates obtained by numerically maximizing the truncated and exact likelihoods using this heuristic for $N_{\max}$ we generated true data from different values of $\lambda$ and $\rho$ with $\lambda\rho = \mathbb{E}[y]$ fixed to be equal to 10—the modeler does not know the true parameters, and in each cases chooses $N_{\max} = 5\mathbb{E}[y] = 50$. Figure 5 shows the results. As the true $\lambda$ increases close to and beyond $N_{\max}$, the truncated method cuts off significant portions of the probability mass and severely underestimates $\lambda$. Estimation with the exact likelihood is noisier as $\lambda$ increases and $\rho \to 0$, but not biased by truncation. While this result is not surprising, it reflects a realistic situation faced by the practitioner who must select this trunctation parameter.

**Marginals.** We demonstrate the computation of posterior marginals and parameter estimation on an end-to-end case study to model the abundance of Northern Dusky Salamanders at 15 sites in the mid-Atlantic US using data from [28]. The data consists of 14 counts at each site, conducted in June and July over 7 years. We first fit a Poisson HMM by numerically maximizing the likelihood as computed by PGF-FORWARD. The model has three parameters total, which are shared across sites and time: arrival rate, survival rate, and detection probability. Arrivals are modeled as a homogenous Poisson process, and survival is modeled by assuming indvidual lifetimes are exponentially distributed. The fitted parameters indicated an arrival rate of 0.32 individuals per month, a mean lifetime of 14.25 months, and detection probability of 0.58.

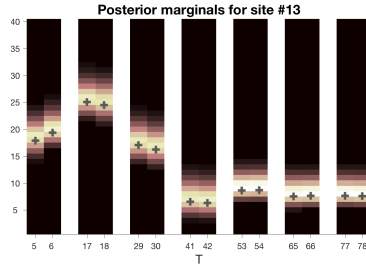

Figure 6: Posterior marginals for abundance of Northern Dusky Salamanders at 1 site. See text.

Figure 6 shows the posterior marginals as computed by PGF-TAIL-ELIMINATE with the fitted parameters, which are useful both for model diagnostics and for population status assessments. The crosses show the posterior mean, and color intensity indicates the actual PMF. Overall, computing maximum likelihood estimates required 189 likelihood evaluations and thus $189 \times 15 = 2835$ calls to PGF-FORWARD, which took 24s total. Extracting posterior marginals at each site required 14 executions of the full PGF-TAIL-ELIMINATE routine (at all 14 latent variables), and took 1.6s per site. Extracting the marginal probabilities and posterior mean took 0.0012s per latent variable.

## 5 Conclusion

We have presented techniques for exact inference in countably infinite latent variable models using probability generating functions. Although many aspects of the methodology are general, the current method is limited to HMMs with Poisson latent variables, for which we can represent and manipulate PGFs efficiently (cf. Theorems 1 and 3). Future work will focus on extending the methods to graphical models with more complex structures and to support a larger set of distributions, for example, including the negative binomial, geometric, and others. One path toward these goals is to find a broader parametric representation for PGFs that can be manipulated efficiently.

**Acknowledgments.** This material is based upon work supported by the National Science Foundation under Grant No. 1617533.

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
