[Supplementary Material]

# A   Proofs — Operations on Generating Functions

*Proof of Proposition 1.*  This is a standard fact about multivariate PGFs:

$$F_\alpha(s_{\alpha\setminus i}, 1) = \sum_{x_\alpha} \psi_\alpha(x_\alpha) s_{\alpha\setminus i}^{x_{\alpha\setminus i}} 1^{x_i} = \sum_{x_{\alpha\setminus i}} \left( \sum_{x_i} \psi_\alpha(x_{\alpha\setminus i}, x_i) \right) s_{\alpha\setminus i}^{x_{\alpha\setminus i}}$$

The fact $\sum_{x_\alpha} \psi_\alpha(x_\alpha) = F_\alpha(1, \dots, 1)$ follows by marginalizing each variable one at a time.   □

*Proof of Proposition 2.*

$$\left. \frac{\partial^a}{\partial s_i^a} F_\alpha(s_\alpha) \right|_{s_i=0} = \sum_{x_{\alpha\setminus i}} \sum_{x_i} \psi_\alpha(x_{\alpha\setminus i}, x_i) s_{\alpha\setminus i}^{x_{\alpha\setminus i}} \left. \frac{\partial^a}{\partial s_i^a} s_i^{x_i} \right|_{s_i=0} = a! \sum_{x_{\alpha\setminus i}} \psi_\alpha(x_{\alpha\setminus i}, a) s_{\alpha\setminus i}^{x_{\alpha\setminus i}}$$

The final equality holds because $\left. \frac{\partial^a}{\partial s_i^a} s_i^{x_i} \right|_{s_i=0} = a!$ if $x_i = a$ and zero otherwise.   □

*Proof of Proposition 3.*  The PGF is

$$\begin{aligned}
F_{\alpha\cup j}(s_\alpha, s_j) &= \sum_{x_\alpha} \sum_{x_j} \psi_\alpha(x_\alpha) \, \mathrm{Binomial}(x_j \mid x_i, \rho) s_\alpha^{x_\alpha} s_j^{x_j} \\
&= \sum_{x_\alpha} \psi_\alpha(x_\alpha) s_\alpha^{x_\alpha} \sum_{x_j} \mathrm{Binomial}(x_j \mid x_i, \rho) s_j^{x_j} \\
&= \sum_{x_\alpha} \psi_\alpha(x_\alpha) s_\alpha^{x_\alpha} (\rho s_j + 1 - \rho)^{x_i} \\
&= \sum_{x_\alpha} \psi_\alpha(x_\alpha) s_{\alpha\setminus i}^{x_{\alpha\setminus i}} \big( s_i(\rho s_j + 1 - \rho) \big)^{x_i} \\
&= F_\alpha(s_{\alpha\setminus i}, s_i(\rho s_j + 1 - \rho))
\end{aligned}$$

In the third line, we used the fact that the PGF of the Binomial distribution is $\sum_x \mathrm{Binomial}(x|n, \rho) s^x = (\rho s + 1 - \rho)^n$.   □

*Proof of Proposition 4.*

$$\begin{aligned}
F_\gamma(s_\alpha, s_\beta, s_k) &= \sum_{x_\alpha, x_\beta, x_k} \psi_\alpha(x_\alpha) \psi_\beta(x_\beta) \mathbb{I}\{x_k = x_i + x_j\} s_\alpha^{x_\alpha} s_\beta^{x_\beta} s_k^{x_k} \\
&= \sum_{x_\alpha, x_\beta} \psi_\alpha(x_\alpha) \psi_\beta(x_\beta) s_\alpha^{x_\alpha} s_\beta^{x_\beta} s_k^{x_i + x_j} \\
&= \sum_{x_\alpha, x_\beta} \psi_\alpha(x_\alpha) \psi_\beta(x_\beta) \cdot s_{\alpha\setminus i}^{x_{\alpha\setminus i}} \cdot (s_k s_i)^{x_i} \cdot s_{\beta\setminus j}^{x_{\beta\setminus j}} \cdot (s_k s_j)^{x_j} \\
&= \left( \sum_{x_\alpha} \psi_\alpha(x_\alpha) \cdot s_{\alpha\setminus i}^{x_{\alpha\setminus i}} \cdot (s_k s_i)^{x_i} \right) \cdot \left( \sum_{x_\beta} \psi_\beta(x_\beta) \cdot s_{\beta\setminus j}^{x_{\beta\setminus j}} \cdot (s_k s_j)^{x_j} \right) \\
&= F_\alpha(s_{\alpha\setminus i}, s_k s_i) \cdot F_\beta(s_{\beta\setminus j}, s_k s_j)
\end{aligned}$$

□

*Proof of Proposition 5.*  We can combine Propositions 3 and 2 to first expand the factor with a thinned variable $x_j = \rho \circ x_i$ and then observe $x_j = a$. We get

$$\begin{aligned}
F_\alpha'(s_\alpha) &= \frac{1}{a!} \left. \frac{\partial^a}{\partial s_j^a} F_\alpha(s_{\alpha\setminus i}, s_i(\rho s_j + 1 - \rho)) \right|_{s_j=0} \\
&= \frac{1}{a!} \left( \left. \frac{\partial^a}{\partial t_i^a} F_\alpha(s_{\alpha\setminus i}, t_i)(s_i \rho)^a \right|_{t_i = s_i(\rho s_j + 1 - \rho)} \right) \Bigg|_{s_j=0} \\
&= \frac{1}{a!} (s_i \rho)^a \left. \frac{\partial^a}{\partial t_i^a} F_\alpha(s_{\alpha\setminus i}, t_i) \right|_{t_i = s_i(1-\rho)}.
\end{aligned}$$

□

**Algorithm 4** PGF-FORWARD implementation

---

**Input:** Vectors $\lambda, \delta, \rho, y$
**Output:** Likelihood $p(y_{1:K})$

1: $a \leftarrow 0, b \leftarrow 0, f(s) \leftarrow 1$
2: **for** $k = 1$ to $K$ **do**
3:     $[a, b] \leftarrow \text{ARRIVALS}(a, b, \lambda_k)$
4:     $[a, f] \leftarrow \text{EVIDENCE}(a, f, y_k, \rho_k)$
5:     **if** $k < K$ **then**
6:         $[a, b, f] \leftarrow \text{SURVIVORS}(a, b, f, \delta_k)$
7:     **end if**
8: **end for**
9: **return** $f(1)\exp\{a + b\}$

10: **function** ARRIVALS$(a, b, \lambda)$
11:     $a' \leftarrow a + \lambda$
12:     $b' \leftarrow b - \lambda$
13:     **return** $a', b'$
14: **end function**

15: **function** EVIDENCE$(a, f, y, \rho)$
16:     $a' \leftarrow a(1 - \rho)$
17:     $g \leftarrow 0, df \leftarrow f$
18:     **for** $\ell = 0$ to $y$ **do**
19:         $g \leftarrow g + df/(a^{\ell}\ell!(y - \ell)!)$
20:         $df \leftarrow \text{DERIV}(df)$
21:     **end for**
22:     $g \leftarrow \text{COMPOSE}(g, s(1 - \rho))$
23:     $g \leftarrow (a\rho)^y s^y g$
24:     **return** $a', g$
25: **end function**

26: **function** SURVIVORS$(a, b, f, \delta)$
27:     $a' \leftarrow a\delta$
28:     $b' \leftarrow b + a(1 - \delta)$
29:     $f' \leftarrow \text{COMPOSE}(f, \delta s + 1 - \delta)$
30:     **return** $a', b', f'$
31: **end function**

---

*Proof of Proposition 6.* This is an immediate consequence of Proposition 3 and Proposition 1 by setting $s_i = 1$ in Proposition 3. $\square$

*Proof of Proposition 7.* This is an immediate consequence of Proposition 4 and Proposition 1 by setting $s_i = 1$ and $s_j = 1$ in Proposition 4. $\square$

## B   Implementation of PGF-FORWARD

The detailed algorithm, based on the proof of Theorem 1, is listed in Algorithm 4.

Here is the proof of the runtime result (Theorem 2):

*Proof of Theorem 2.* We assume a polynomial $f$ is represented as a vector of coefficients $\{f_i\}$ of length $\deg(f) + 1$. ARRIVALS takes constant time. The running time of EVIDENCE is $\mathcal{O}(y \deg(f)) = \mathcal{O}(Y^2)$: Lines 19 and 20 are executed $y$ times and take time proportional to $\deg(g)$ and $\deg(df)$, respectively, each of which is no more than $\deg(f)$. The operations outside the loop are bounded by $\mathcal{O}(y + \deg(f))$. (Note that the COMPOSE operation in Line 22 is linear in $\deg(g)$—simply multiply the $i$th coefficient of $g$ by $(1 - \rho)^i$ for all $i$.) The SURVIVORS function takes $\mathcal{O}(Y^2)$ time. The COMPOSE operation in Line 29 is more costly than the one on Line 22: we must expand $\sum_i g_i(\delta s + 1 - \delta)^i$ to compute the coeffients of $s^i$ for all $i$—this can be done in $O(\deg(g)^2)$ time by a number of methods, e.g., applying the Binomial Theorem to expand each term. The ARRIVALS, EVIDENCE, and SURVIVORS functions are each called $K$ or $K - 1$ times. Therefore, the overall running time is $\mathcal{O}(KY^2)$. $\square$

## C   Implementation of PGF-TAIL-ELIMINATE

We provide a side-by-side comparison of PGF-TAIL-ELIMINATE with a non-PGF implementation of the equivalent algorithm, TAIL-ELIMINATE, in Figure 7. The detailed PGF-TAIL-ELIMINATE algorithm is listed in Algorithm 7.

*Proof of Theorem 3.* We again proceed inductively. From the proof of Theorem 1, we initially have that $A_i(s) = f(s)\exp\{as + b\}$ where $\deg(f) = \sum_{k=1}^{i} y_k$. Then, in Line 1, we have

$$\Psi_{i,i+1}(s, t) = f\big(s(\delta_i t + 1 - \delta_i)\big)\exp\{a\delta_i st + a(1 - \delta_i)s + b\}$$

The first term is a bivariate polynomial $f'(s, t) := \sum_{i=0}^{\deg(f)} f_i s^i (\delta_i t + 1 - \delta_i)^i$ with max-degree equal to $\deg(f)$, and the second term has the desired exponential form.

| **Algorithm 5** TAIL-ELIMINATE | **Algorithm 6** PGF-TAIL-ELIMINATE |
|---|---|
| **Output:** Unnormalized marginal $p(n_i, y_{1:K})$ | **Output:** PGF of unnormalized marginal $p(n_i, y_{1:K})$ |
| 1: $\phi_{i,i+1}(n_i, z_{i+1}) := \alpha_i(n_i)p(z_{i+1}\|n_i)$ | 1: $\Phi_{i,i+1}(s,t) := A_i(s(\delta_i t + 1 - \delta_i))$ |
| 2: **for** $j = i+1$ to $K$ **do** | 2: **for** $j = i+1$ to $K$ **do** |
| 3: $\quad \eta_{ij}(n_i, n_j) := \sum_{m_j, z_j} \phi(n_i, z_j)p(m_j)p(n_j\|z_j, m_j)$ | 3: $\quad H_{ij}(s,t) := \Phi_{ij}(s,t)\exp\{\lambda_k(t-1)\}$ |
| 4: $\quad \theta_{ij}(n_i, n_j) := \eta_{ij}(n_i, n_j)p(y_j\|n_j)$ | 4: $\quad \Theta_{ij}(s,t) := \frac{1}{y_j!}(t\rho_j)^{y_j} \frac{\partial^{y_j} H_{ij}(s,u)}{\partial u^{y_j}}\Big\|_{u=t(1-\rho_j)}$ |
| 5: $\quad$ **if** $j < K$ **then** | 5: $\quad$ **if** $j < K$ **then** |
| 6: $\quad\quad \phi_{i,j+1}(n_i, z_{j+1}) := \theta_{ij}(n_i, n_j)p(z_j\|n_{j-1})$ | 6: $\quad\quad \Phi_{i,j+1}(s,t) := \Theta_{ij}(s, \delta_j t + 1 - \delta_j)$ |
| 7: $\quad$ **end if** | 7: $\quad$ **end if** |
| 8: **end for** | 8: **end for** |
| 9: **return** $p(n_i, y_{1:K}) = \sum_{n_K} \theta_{iK}(n_i, n_K)$ | 9: **return** $\Theta_{iK}(s, 1)$ |

Figure 7: Comparison of the PGF-TAIL-ELIMINATE algorithm with its equivalent using non-PGF factors, TAIL-ELIMINATE.

In Line 3, suppose $\Phi_{ij}(s,t) = f(s,t)\exp\{ast + bs + ct + d\}$. Then $H_{ij}(s,t) = f(s,t)\exp\{ast + cs + (c+\lambda_k)t + (d-\lambda_k)\}$, which has the desired form.

In Line 4, the suppose $H_{ij}(s,u) = f(s,u)\exp\{ast + bs + cu + d\}$. One can verify by calculating the $y$th partial derivative of $H_{ij}$ with respect to $u$ that:

$$\Theta_{ij}(s,t) = \rho_j^{y_j} \cdot \left( t^{y_j} \sum_{\ell=0}^{y_j} \frac{(as+c)^{y_j-\ell}}{\ell!(y_j-\ell)!} \cdot \frac{\partial^\ell}{\partial u^\ell} f(s,u)\Big|_{u=t(1-\rho_j)} \right) \cdot \exp\left\{a(1-\rho_j)st + bs + c(1-\rho)t + d\right\}$$

The term in parentheses is again a bivariate polynomial—the largest exponent of $s$ and $t$ have both increased by $y_j$, so the max-degree increases by $y_j$. The exponential term is in the desired form and can absorb the scalar $\rho^{y_j}$. Therefore, in Line 4, $\Theta_{ij}(s,t)$ has the desired form, and the degree of the polynomial part of the representation increases by $y_j$.

In Line 6, suppose $\Theta_{ij}(s,t) = f(s,t)\exp\{ast + bs + ct + d\}$. Then $\Phi_{i,j+1}(s,t) = g(s,t)\exp\left\{a\delta_k st + (b + a(1-\delta_k))s + c\delta_k t + (d + c(1-\delta_k))\right\}$, where $g(s,t) = f(s, h(t))$ is the composition of $f$ with the affine function $h(t) = \delta_k t + 1 - \delta_k$, so $g$ is a bivariate polynomial of the same degree as $f$. Therefore, $\Phi_{i,j+1}(s,t)$ has the desired form.

We have shown that each PGF retains the desired form. Furthermore, the max-degree of the polynomial is initially equal to $\sum_{k=1}^{i} y_k$ and increases by $y_j$ for all $j = i+1$ to $K$, so it is always bounded by $Y = \sum_{k=1}^{K} y_k$. $\qquad\square$

*Proof of Theorem 4 (*PGF-TAIL-ELIMINATE *running time).* We assume for simplicity that all polynomials have max-degree equal to the upper bound $Y$. A bivariate polynomial is represented as a matrix of $Y^2$ coefficients for the monomials $s^i t^j$.

The running time of INIT-SURVIVORS function is dominated by Line 16, which takes $\mathcal{O}(Y^2)$ time. For each term in the sum, the coefficients of the polynomial $(\delta t + 1 - \delta)^i$ can be computed in $\mathcal{O}(i) = \mathcal{O}(Y)$ time (e.g., by the Binomial Theorem) and then multiplied by $f_i$ to determine the coefficients of $s^i t^j$ for all $j$. This repeats $\mathcal{O}(Y)$ times, once for each term in the sum.

The running time of ARRIVALS is $\mathcal{O}(1)$.

The running time of SURVIVORS is $\mathcal{O}(Y^3)$. The COMPOSE operation in Line 41 can be structured as

$$\sum_{i,j} f_{ij} s^i (\delta t + 1 - \delta)^j = \sum_i s^i \sum_j f_{ij}(\delta t + 1 - \delta)^j$$

For each value of $i$, we compose the univariate polynomial $\sum_j f_{ij} t^j$ with the affine function $\delta t + 1 - \delta$. This can be done in $O(Y^2)$ time, as in the proof of Theorem 2, for a total running time of $O(Y^3)$. $\qquad\square$

**Algorithm 7** PGF-TAIL-ELIMINATE implementation

---

**Input:** Vectors $\lambda, \delta, \rho, y$, index $i$, parameters $f, a, b$ of initial PGF $A_i(s) = f(s)\exp\{as + b\}$ (from PGF-FORWARD)
**Output:** Final PGF for unnormalized marginal $p(n_i, y_{1:K})$ in form $f(s)\exp\{as + b\}$

1: // Initialize: $f(s, t)\exp\{ast + bs + ct + d\}$
2: $[a, b, c, d, f] \leftarrow$ INIT-SURVIVORS$(a, b, f, \delta_i)$
3: **for** $j = i + 1$ to $K$ **do**
4:     $[c, d] \leftarrow$ ARRIVALS$(c, d, \lambda_k)$
5:     $[a, c, f] \leftarrow$ EVIDENCE$(a, c, f, y_k, \rho_k)$
6:     **if** $k < K$ **then**
7:         $[a, b, c, d, f] \leftarrow$ SURVIVORS$(a, b, c, d, f, \delta_k)$
8:     **end if**
9: **end for**
10: **return** $f(s, 1)\exp\{(a + b)s + (c + d)\}$

11: **function** INIT-SURVIVORS$(a, b, f, \delta)$
12:     $a' \leftarrow a\delta$
13:     $b' \leftarrow b(1 - \delta)$
14:     $c' \leftarrow 0$
15:     $d' \leftarrow b$
16:     $f'(s, t) \leftarrow \sum_i f_i s^i (\delta t + 1 - \delta)^i$
17:     **return** $a', b', c', d', f'$
18: **end function**

19: **function** ARRIVALS$(c, d, \lambda)$
20:     $c' \leftarrow c + \lambda$
21:     $d' \leftarrow d - \lambda$
22:     **return** $c', d'$
23: **end function**

24: **function** EVIDENCE$(a, c, f, y, \rho)$
25:     $a' \leftarrow a(1 - \rho)$
26:     $c' \leftarrow c(1 - \rho)$
27:     $g \leftarrow 0, df \leftarrow f$
28:     **for** $\ell = 0$ to $y$ **do**
29:         $g \leftarrow g + \dfrac{\text{MULT}(df, (as + c)^{y-\ell})}{\ell!(y - \ell)!}$
30:         $df \leftarrow$ PARTIAL$(df, t)$
31:     **end for**
32:     $g \leftarrow$ COMPOSE$(g, t(1 - \rho))$
33:     $g \leftarrow \rho^y s^y g$
34:     **return** $a', g$
35: **end function**

36: **function** SURVIVORS$(a, b, f, \delta)$
37:     $a' \leftarrow a\delta$
38:     $b' \leftarrow b + a(1 - \delta)$
39:     $c' \leftarrow c\delta$
40:     $d' \leftarrow d + c(1 - \delta)$
41:     $f' \leftarrow$ COMPOSE$(f, \delta t + 1 - \delta)$
42:     **return** $a', b', f'$
43: **end function**

---

The total running time of PGF-TAIL-ELIMINATE *excluding* the EVIDENCE function is therefore $\mathcal{O}(KY^3)$.

The running time of one call to EVIDENCE is $\mathcal{O}(yY^2 \log Y)$. It is dominated by Line 29. The multiplication in this line can be structured as

$$\left(\sum_{i,j}(df)_{ij}s^i t^j\right)(as + c)^{y-\ell} = \sum_j t^j \left(\sum_i (df)_{ij}s^i\right)(as + c)^{y-\ell}$$

For each value of $j$, we multiply two univariate polynomials in $s$ whose total degree is at most $Y$. This can be done in time $\mathcal{O}(Y \log Y)$ using a fast Fourier transform. We repeat this at most $Y \cdot y$ times—once for each possible value of $j$ and $\ell$. The total running time of a single call to EVIDENCE is therefore $\mathcal{O}(yY^2 \log Y)$. The total running time of all calls to the evidence function is $\mathcal{O}(\sum_{j=i+1}^K y_k Y^2 \log Y) = \mathcal{O}(Y^3 \log Y)$.

The overall running time is therefore $\mathcal{O}(Y^3(K + \log Y))$.

# D   Proof of Theorem 5 — Extracting Marginal Probabilities and Moments

*Proof of Theorem 5.* We assume for the proof that the PGF is already normalized, which can be done by setting $b \leftarrow b - \log Z$. For (i) and (ii), we use the following standard facts about PGFs:

$\mu = F^{(1)}(1)$ and $\sigma^2 = F^{(2)}(1) - \mu^2 + \mu$ [3]. Then we have:

$$\mu = F^{(1)}(1) = \frac{d}{ds} f(s) e^{as+b}\Big|_{s=1}$$
$$= f'(1)e^{a+b} + af(1)e^{a+b}$$
$$= e^{a+b} \sum_{i=0}^{m} (mf_i + af_i)$$
$$= e^{a+b} \sum_{i=0}^{m} (a+m)f_i$$

And

$$F^{(2)}(1) = \frac{d^2}{ds^2} f(s) e^{as+b}\Big|_{s=1}$$
$$= f^{(2)}(1)e^{a+b} + 2f^{(1)}(1)ae^{a+b} + a^2 f(1)$$
$$= e^{a+b} \left( f^{(2)}(1) + 2af^{(1)}(1) + a^2 f(1) \right)$$
$$= e^{a+b} \sum_{i=0}^{m} \left( m(m-1)f_i + 2amf_i + a^2 f_i \right)$$
$$= e^{a+b} \sum_{i=0}^{m} ((a+m)^2 - m)f_i$$

For part (iii), we use the following standard fact about the Taylor expansion of the exponential:

$$e^{as} = \sum_{j=0}^{\infty} \frac{a^j}{j!} s^j$$

Then we have:

$$F(s) = \left( \sum_{i=0}^{m} f_i s^i \right) e^{as+b}$$
$$= e^b \left( \sum_{i=0}^{m} f_i s^i \right) \left( \sum_{j=0}^{\infty} \frac{a^j}{j!} s^j \right)$$
$$= e^b \sum_{i=0}^{m} \sum_{j=0}^{\infty} f_i \frac{a^j}{j!} s^{i+j}$$
$$= e^b \sum_{\ell=0}^{\infty} s^\ell \sum_{i=0}^{\min\{m,\ell\}} f_i \frac{a^{\ell-i}}{(\ell-i)!}$$

The final expression reveals the unique explicit representation of the PGF as a formal power series in $s$. The coefficient of $s^\ell$, which is equal to the value of the PMF at $\ell$, is $e^b \sum_{i=0}^{\min\{m,\ell\}} f_i \frac{a^{\ell-i}}{(\ell-i)!}$.

$\square$