[Reviews · NeurIPS 2016]

Reviewer 1

Summary

This paper proposes a symbolic representation of the intermediate factors generated during variable elimination (VE) in directed graphical models with Poisson and Binomial distributions (over countably infinite state spaces). The representation is that of the probability generating function (PGF) and they show that for certain types of interactions, this PGF remains in the form "polynomial function times exponential of linear function." For a particular chain structure (a form of an HMM), the polynomial's degree remains bounded (by the total sum of the observations). Thus, exact VE can be done in bounded space and time.

Qualitative Assessment

This is a nice result. It isn't clear exactly how general this is. Any distribution can be represented as a PGF. The trick is that for the types of factors in the particular HMMs discussed in this paper, the resulting symbolic forms can be worked out and show to be bounded (in some way). It also isn't clear why the original distributions (instead of the PGF transformations) could not have been treated this way. One could run VE using a symbolic math package in a very general way. This paper shows that the scope necessary for this symbolic representation is limited and thus it is feasible in some situations. ---------- If the PGF description of the intermediate factors from VE has a symbolic form with bounded representation size, does this mean that the symbolic description of the "raw" factor does as well? Why not use that? What is f_i in Theorem 5? In Theorem 5, i should not be used both as the unquantified index of the time index of question (n_i, for instance) and the quantified index of the sum. In Figure 5, why is the exact method so noisy? The under-estimation shown by the truncation method is understandable. However, it isn't clear why the exact method doesn't perform well. Is this because the data sample is small, so the correct ML estimate is far from the true parameter value that generated the data? If so, perhaps the experiment should be repeated enough times to show that the method is unbiased and what std dev is to be expected from both methods. Why is the runtime of the marginal calculation, compared to that of truncation, not shown?

Confidence in this Review

2-Confident (read it all; understood it all reasonably well)


Reviewer 2

Summary

This paper shows how certain types of graphical models with Poisson variables admit exact inference even though the Poisson support is infinite, using the pgf as a finite-dimensional representation.

Qualitative Assessment

This paper was in fact a pleasure to read. The introduction is a cogent summary of why some distributions are harder to write inference algorithms for. The notation is clear and the component terms of equations well explained. The idea of using a closed family of functions induced by pgfs for representing intermediate factors in inference is novel (as far as there is ever anything new under the sun). I am unaware of any PGM literature doing this in general inference algorithm context. I expect this paper to be impactful as this trick is added to the quiver of other people trying to do inference in simple but intractable models. One flaw of the paper is that it does not look forward much to extending this to all exponential family distributionsm, and does not discuss what happens when Poisson variables are mixed in with discrete or Gaussian ones. (Obviously, there are hard problems there analogous to "continuous parents of discrete children"). In general factor multiplication, does the f(s) exp(...) functional form break? Does the representationa dimensionality stop being manageable very fast? In the equation in Section 3.3, it took me a bit to figure out the variable whose values m_j and z_j represented, respectively. In Theorem 5, what is f_i? Is it the polynomial at I-th time step? The numbers on the axes in Figure 4 are hard to read in print. Can you use a bigger font for just the numbers? How do the fitted values from the "Marginals" section of experiments compare to what biologists know of salamanders?

Confidence in this Review

3-Expert (read the paper in detail, know the area, quite certain of my opinion)


Reviewer 3

Summary

This paper extends the standard variable elimination operations for factors to operate on a generating function representation of a factors over infinite discrete random variables. An efficient algorithm is derived for a specific instance.

Qualitative Assessment

I enjoyed reading this paper, mostly because the motivation makes so much sense. We really need this type of basic algorithm for integer-valued random variables. Some more discussion of the limitations could be helpful (why do you get an efficient instantiation for your filtering problem and not for other cases). Also a small worked-out example would be great. What is the connection to collective graphical models?

Confidence in this Review

2-Confident (read it all; understood it all reasonably well)


Reviewer 4

Summary

This paper generalizes the classical variable elimination algorithm in latent variable models from latent variables with finite discrete values or Gaussian distribution to latent variables with countably infinite discrete values, in particular the Poisson latent variable models. The authors observe that the key property that enables fast and exact inference is that operations like conditioning, marginalization and multiplication are "conjugate", producing exact and compact representation. For models with countably infinite factors, they appeal to probability generating function to achieve analogous "conjugacy" so that these operations boast exact and compact representation. This enables variable elimination in such models.

Qualitative Assessment

The idea of probability generating function achieving similar forms of "conjugacy" as Gaussian random variables and (finite) discrete random variables is very interesting. This enables a fast and exact inference algorithm on Poisson latent variable models. The formulation involving probability generating function does not seem to be constrained to Poisson random variables but all the simulations and real application pertain to Poisson HMM. It is unclear if there's anything special with Poisson that enables better parameter estimates. It is also unclear if there are other Poisson latent variable models other than Poisson HMMs with wide applications. Would be great to have this question answered in the article. The comparison of the proposal, in the simulation experiments, is only with truncation in performance and with MCMC in speed. Truncation is known to perform bad when the rate is above the truncation threshold. It would be considered less than complete if it is not compared with MCMC in accuracy. MCMC is known to be particularly slow in convergence. Comparison of computational cost might as well be with truncation and variational inference. By the way, MCMC is also exact but only with sampling -- it is not approximate. The real experiment is not too satisfying as well. It does not compare with either MCMC or truncation or present any new findings with this new proposal. Thus the merit of this proposal is unclear in real experiments as well. Apart from parameter estimation accuracy, it would also be very interesting to see predictive accuracy in real experiments because all models are wrong and it's interesting to see if some are useful.

Confidence in this Review

2-Confident (read it all; understood it all reasonably well)


Reviewer 5

Summary

In this work, the authors propose a variable elimination approach for inference in a model class with unobserved count variables. The approach is based on probability generating functions; the authors define the relevant operations (marginalization, etc.) and give efficient implementations of them. Experiments suggest that the approach outperforms existing techniques like truncation.

Qualitative Assessment

The scope of the contribution seems limited to Poisson hidden variables. For example, it is not clear if the proposed PGF technique could generalize to all exponential families. It is not clear which of the operations on PGFs are novel (if any). It is not clear if the operations described in Propositions 5, 6 and 7 are somehow more efficient than application of the basic operations. It is not clear if the estimated parameters are biologically-sensible. --- Minor comments, typos, etc. Figure 1 (a) could show hyperparameters. The paper does not include any sort of conclusions.

Confidence in this Review

1-Less confident (might not have understood significant parts)


Reviewer 6

Summary

This paper focuses on inference problems related to count data processes based on binomial thinning, referred to as Poisson HMM in the paper. Algorithms based on probability generating function(pgf) are shown to compute likelihood and posterior marginals in pesudo polynomial time. The authors propose based on this example that pgf can be a good representation of factors in graphical models and may solve other intractable classes as well; however this is not demonstrated with any other class of problems, the results in the paper are only on the so called Poisson HMM model.

Qualitative Assessment

The Poisson HMM model, which is the focus of the paper, is actually well known as binomial thinning in the literature on count data time-series, as noted in the paper as well. The poisson assumption seems un-necessary though, as Propositions 1-7 don't rely on Poisson distribution anywhere. However, its surprising that the authors aren't aware that it is already a standard practice to use pgf to represent distributions in this model, because of their ease in working with binomial thinning, as evidenced by Props. 3 and 6 of the paper. In this literature, however it is more common to work with apgf which is E[(1-s)^X], since it leads to simpler expression in Prop. 6, which is the most common operation used. This glaring omission brings into question most of the claims made of novel contributions in this paper. The use of pgf for this model is not new, and the theorems/algorithms proposed seem like they are not hard to derive based on existing work. I may be wrong here, but unless authors clarify what is novel in the paper and distinguish themselves from closely related work which they seem unaware of, its hard to judge the paper. Please see McKenzie, Eddie. "Ch. 16. Discrete variate time series." Handbook of statistics 21 (2003): 573-606 for a survey. Other minor points: Instead of saying "pesudo polynomial time", it may be better to clarify how running time depends on the value of evidence. typo in Sec. 1 last para: *estimatation Parameter estimation is mentioned many times, but the exact algorithm/optimization used is never mentioned.

Confidence in this Review

2-Confident (read it all; understood it all reasonably well)